# Chemotherapy for Biliary Tract Cancer in 2021

**DOI:** 10.3390/jcm10143108

**Published:** 2021-07-14

**Authors:** Takashi Sasaki, Tsuyoshi Takeda, Takeshi Okamoto, Masato Ozaka, Naoki Sasahira

**Affiliations:** Department of Hepato-Biliary-Pancreatic Medicine, Cancer Institute Hospital of Japanese Foundation for Cancer Research, 3-8-31, Ariake, Koto-ku, Tokyo 135-8550, Japan; tsuyoshi.takeda@jfcr.or.jp (T.T.); takeshi.okamoto@jfcr.or.jp (T.O.); masato.ozaka@jfcr.or.jp (M.O.); naoki.sasahira@jfcr.or.jp (N.S.)

**Keywords:** biliary tract cancer, cholangiocarcinoma, chemotherapy, cytotoxic agents, molecular targeted agents, immunotherapy, precision medicine, genetic testing

## Abstract

Biliary tract cancer refers to a group of malignancies including cholangiocarcinoma, gallbladder cancer, and ampullary cancer. While surgical resection is considered the only curative treatment, postoperative recurrence can sometimes occur. Adjuvant chemotherapy is used to prolong prognosis in some cases. Many unresectable cases are also treated with chemotherapy. Therefore, systemic chemotherapy is widely introduced for the treatment of biliary tract cancer. Evidence on chemotherapy for biliary tract cancer is recently on the increase. Combination chemotherapy with gemcitabine and cisplatin is currently the standard of care for first-line chemotherapy in advanced cases. Recently, FOLFOX also demonstrated efficacy as a second-line treatment. In addition, efficacies of isocitrate dehydrogenase inhibitors and fibroblast growth factor receptor inhibitors have been shown. In the adjuvant setting, capecitabine monotherapy has become the standard of care in Western countries. In addition to conventional cytotoxic agents, molecular-targeted agents and immunotherapy have been evaluated in multiple clinical trials. Genetic testing is used to check for genetic alterations and molecular-targeted agents and immunotherapy are introduced based on tumor characteristics. In this article, we review the latest evidence of chemotherapy for biliary tract cancer.

## 1. Introduction

Biliary tract cancer is a heterogeneous group of highly aggressive cancers including intrahepatic/perihilar/distal cholangiocarcinoma, gallbladder cancer, and ampullary cancer [1]. Biliary tract cancer is common in Japan, Southeast Asia, South America, and India [2,3]. Cholangiocarcinoma has been increasing worldwide, while the incidence of gallbladder cancer has been decreasing in recent years [4,5,6]. In Japan, the incidence and mortality of biliary tract cancer have plateaued over the last decade, with an annual incidence and mortality of approximately 22,000 and 18,000, respectively [7]. This cancer is still the sixth leading cause of cancer-related death. In Japan, more than 45% of new cases are diagnosed over the age of 80.

While surgical resection is considered the only curative treatment, postoperative recurrence can sometimes occur. Data from the biliary tract cancer registry in Japan revealed that five-year survival rates were 39.8% for gallbladder cancer, 24.2% for perihilar cholangiocarcinoma, 39.1% for distal cholangiocarcinoma, and 61.3% for ampullary cancer [8]. Adjuvant chemotherapy is sometimes introduced to achieve long-term survival for resected cases with poor prognostic factors. Many unresectable cases are also treated with chemotherapy. As surgery for biliary tract cancer is a highly invasive procedure, surgery may be avoided in potentially resectable cases due to old age or comorbidities. Therefore, systemic chemotherapy is widely introduced for the treatment of biliary tract cancer. Recently, evidence on chemotherapy for biliary tract cancer is on the increase. In addition to conventional cytotoxic agents, molecular-targeted agents and immunotherapy have widely been introduced in this field. Genetic testing is used to check for genetic alterations and molecular-targeted agents and immunotherapy are introduced based on tumor characteristics. Here, we review the latest evidence on chemotherapy for biliary tract cancer.

## 2. First-Line Chemotherapy for Advanced Biliary Tract Cancer

Standard chemotherapy for biliary tract cancer was not established until about 2000. Until then, chemotherapy for pancreatic cancer had been used as a reference. The efficacy of chemotherapy was confirmed in a randomized control study conducted before 2000 which compared chemotherapy to best supportive care in advanced pancreatic and biliary tract cancers [9]. Subsequently, a randomized controlled study comparing chemotherapy and best supportive care for unresectable gallbladder cancer was reported from India in 2010, confirming the usefulness of chemotherapy [10]. Between 2000 and 2010, gemcitabine and 5-fluorouracil were considered the key drugs for the treatment of advanced cases. A pooled analysis of clinical trials conducted between 1985 and 2006 identified gemcitabine, fluoropyrimidines, and cisplatin as the key active agents and concluded that gemcitabine combined with platinum compounds represented the provisional standard of chemotherapy for advanced biliary tract cancer [11].

The combination chemotherapy of gemcitabine and platinum compounds demonstrated good efficacy in advanced cases. A randomized phase II study (ABC-01) comparing the doublet of gemcitabine and cisplatin to gemcitabine alone was reported from the United Kingdom [12]. The doublet regimen was associated with improved tumor control and progression-free survival. Based on this result, the study was extended to a phase III study (ABC-02) to verify the prognostic effect of the combination chemotherapy relative to gemcitabine monotherapy [13]. Four hundred ten patients were randomized to receive either gemcitabine and cisplatin combination chemotherapy or gemcitabine alone. The primary endpoint was overall survival. The median overall survival was 11.7 months in the combination group and 8.1 months in the monotherapy group (hazard ratio, 0.64; *p <* 0.001). The median progression-free survivals of the combination and monotherapy groups were 8.0 months and 5.0 months, respectively (*p <* 0.001). The rate of tumor control among patients in the combination group was significantly increased (81.4% vs. 71.8%, *p =* 0.049). Although neutropenia occurred more frequently in the combination group, combination chemotherapy with gemcitabine and cisplatin was considered a feasible regimen for advanced biliary tract cancer. This combination chemotherapy was also evaluated in Japanese patients and similar efficacy was confirmed in a multicenter, randomized phase II study (BT-22) [14]. Treatment was repeated for up to 24 weeks in the ABC-02 study and up to 48 weeks in the BT-22 study. In a meta-analysis of these two studies, the efficacy of gemcitabine and cisplatin combination chemotherapy was confirmed in patients with good performance status (performance status of 0 or 1) and in patients with cholangiocarcinoma or gallbladder cancer [15]. On the other hand, the superiority of this combination chemotherapy was not shown in patients with poor performance status or ampullary cancer. The major grade 3/4 adverse events of gemcitabine and cisplatin combination chemotherapy were neutropenia and anemia. We also need to pay attention to renal dysfunction and hearing loss. Oxaliplatin is another platinum compound known to cause less renal damage and therefore does not require aggressive hydration, unlike cisplatin. Oxaliplatin is sometimes used as a substitute for cisplatin. However, the non-inferiority of gemcitabine and oxaliplatin combination chemotherapy, when compared to gemcitabine and cisplatin combination chemotherapy, has not been proven. One randomized controlled study comparing these two regimens was conducted in India [16]. A total of 243 patients with unresectable gallbladder cancer were randomly assigned to one of these two regimens. The median overall survivals of gemcitabine and oxaliplatin combination chemotherapy and gemcitabine and cisplatin combination chemotherapy were 9.0 months and 8.3 months, respectively (hazard ratio, 0.78; *p =* 0.057). Because the predetermined statistical threshold was not met, the study failed to prove non-inferiority. Moreover, this study was underpowered to determine the superiority of gemcitabine and oxaliplatin combination chemotherapy.

Several randomized controlled studies have been conducted in pursuit of treatment regimens that are superior to the standard treatment of gemcitabine and platinum compounds. Some involved combination chemotherapies which added a third drug to the doublet, while others involved a novel regimen. Table 1 summarizes previous randomized controlled studies on first-line chemotherapy for advanced biliary tract cancer. No additional benefits of epidermal growth factor receptor and vascular endothelial growth factor receptor inhibitors have been observed to date [17]. On the other hand, good results have been obtained with S-1, which is widely used in Japan [18,19].

S-1 is an oral fluoropyrimidine derivative used mainly in Asian countries. The combination of gemcitabine and S-1 was widely evaluated in phase II and randomized phase II studies in Japan [34,35,36,37]. Based on these results, a randomized phase III study comparing gemcitabine and S-1 combination chemotherapy with gemcitabine and cisplatin combination chemotherapy was conducted in Japan [18]. This study was conducted to evaluate the non-inferiority of gemcitabine and S-1 combination chemotherapy compared to gemcitabine and cisplatin combination chemotherapy. Patients with advanced biliary tract cancer were randomly assigned either gemcitabine and S-1 combination chemotherapy or gemcitabine and cisplatin combination chemotherapy. The primary endpoint was overall survival. The median overall survivals were 15.1 months and 13.4 months, respectively (hazard ratio 0.945, *p* = 0.046 for non-inferiority). Because the toxicities of gemcitabine and S-1 combination chemotherapy were deemed acceptable, this new doublet also became the standard of care for patients with advanced biliary tract cancer. The major grade 3/4 adverse event of gemcitabine and S-1 combination chemotherapy was neutropenia. We also need to pay attention to diarrhea, oral mucositis, maculopapular rash, and skin hyperpigmentation. S-1 was also evaluated as the triplet with gemcitabine and cisplatin. Based on the good result of a phase II study evaluating the efficacy of gemcitabine + cisplatin + S-1 combination chemotherapy [38], a phase III study was conducted to confirm the superiority of this triplet over gemcitabine and cisplatin combination chemotherapy in Japan [19]. Two hundred forty-six patients with advanced biliary tract cancer were randomized 1:1 to receive either the triplet or the doublet chemotherapy. The primary endpoint was overall survival. The median overall survivals of gemcitabine + cisplatin + S-1 combination chemotherapy and gemcitabine + cisplatin combination chemotherapy were 13.5 months and 12.6 months, respectively (hazard ratio 0.791, *p* = 0.046). This adverse event’s profile of the triplet chemotherapy was also acceptable. The major grade 3/4 adverse event of triplet chemotherapy was also neutropenia. This triplet is also needed to pay attention to diarrhea, stomatitis, and rash. Therefore, gemcitabine + cisplatin + S-1 combination chemotherapy is currently considered a standard regimen for advanced cases.

In summary, the global standard first-line chemotherapy for advanced biliary tract cancer is still gemcitabine and cisplatin combination chemotherapy. In Japan, gemcitabine + S-1 combination chemotherapy and gemcitabine + cisplatin + S-1 combination chemotherapy are also considered alternatives of gemcitabine + cisplatin combination chemotherapy in the first-line setting.

## 3. Second-Line Chemotherapy for Advanced Biliary Tract Cancer

The usefulness of second-line chemotherapy has been reported based on a systematic review and large retrospective studies, but standard treatment has not been established [39,40,41,42,43,44,45]. In Japan, S-1 is widely used as monotherapy in the clinical setting [46,47]. To establish the standard treatment of second-line chemotherapy, various treatments such as molecular-targeted agents and immunotherapy are being developed in addition to conventional cytotoxic agents [48]. Recently, several randomized phase II and phase III studies were reported, some of which showed positive results. Table 2 summarizes previous randomized controlled studies of second-line or third-line chemotherapy for advanced biliary tract cancer.

A phase III study (ABC-06) comparing FOLFOX (5-fluorouracil + leucovorin + oxaliplatin) and active symptom control was conducted in the United Kingdom [50]. Patients with advanced biliary tract cancer treated previously with gemcitabine and cisplatin combination chemotherapy were included. Enrolled patients were randomized to receive either FOLFOX or active symptom control, which was the equivalent of best supportive care. Patients in the active symptom control group could receive FOLFOX after radiographic disease progression was confirmed. The primary endpoint was overall survival. The median overall survivals of FOLFOX and active symptom control groups were 6.2 months and 5.3 months, respectively (hazard ratio 0.69, *p* = 0.031). The benefit of FOLFOX was consistent across subgroups, including those with platinum sensitivity during first-line chemotherapy. The major grade 3/4 adverse events of FOLFOX were neutropenia, fatigue, and catheter-related infection. We also need to pay attention to peripheral neuropathy. This study was the first prospective phase III study that confirmed the benefit of chemotherapy after combination chemotherapy with gemcitabine and cisplatin. Another positive phase III study that showed the efficacy of second-line chemotherapy was the ClarIDHy study. This study was a global phase III study comparing ivosidenib and best supportive care. Ivosidenib is a first-in-class, oral, targeted, small-molecule inhibitor of mutant isocitrate dehydrogenase (IDH) 1 protein. IDH1 mutations occur in up to 20% of cholangiocarcinomas. Patients with advanced cholangiocarcinoma who had received 1–2 prior therapies were enrolled in this study. Patients were randomly assigned to either the ivosidenib group or the best supportive care group. The primary endpoint was progression-free survival. The median progression-free survivals of the ivosidenib and best supportive care groups were 2.7 months and 1.4 months, respectively (hazard ratio 0.37, *p* < 0.001). The major grade 3/4 adverse events of ivosidenib were reported as ascites. This study was the first prospective phase III study that demonstrated a clinical benefit in targeting a molecularly defined subgroup of cholangiocarcinoma and in evaluating genetic profiles of biliary tract cancer. In 2021, the result of a randomized phase II study (NIFTY) comparing 5-fluorouracil + leucovorin + nano-liposomal irinotecan and 5-fluorouracil + leucovorin was reported [58]. This triplet chemotherapy is now known as the NAPOLI regimen and is widely used for second-line chemotherapy in advanced pancreatic cancer. The additional benefit of nano-liposomal irinotecan was demonstrated in this study. Two other randomized phase II studies also showed positive results with capecitabine and irinotecan combination chemotherapy and with regorafenib monotherapy. However, the number of patients enrolled in these studies was relatively small. Therefore, further evaluation is required to establish more solid evidence on these two regimens.

Biliary tract cancers are a heterogeneous group of cancers with different genetic alteration profiles [59,60,61,62]. Potential clinically actionable alterations, defined as oncogenic driver alterations with matched therapeutic agents either under investigation or approved in other tumor types, were identified in 44.5% of patients, showing promise for precision medicine in this field [62]. Common genes implicated in biliary tract cancer tumorigenesis include IDH1, IDH2, fibroblast growth factor receptor (FGFR) 1, FGFR2, FGFR3, and human epidermal growth factor receptor (HER) 2. Encouraging results were seen in patients with identified mutational targets, especially in tumors harboring FGFR2 fusions, HER2, and IDH mutations. The efficacy of an IDH1 inhibitor (ivosidenib) was shown in a phase III study [49]. Several FGFR inhibitors have been evaluated in phase II studies [63,64,65,66]. FGFR2 rearrangements were reported in 7.4% and 3.6% of Japanese intrahepatic cholangiocarcinoma and perihilar cholangiocarcinoma patients, respectively [67]. Based on the results of a phase II study (FIGHT-202) [65], pemigatinib was approved in many countries for patients with FGFR2 fusion or rearrangement. The major grade 3/4 adverse events of pemigatinib were hypophosphatemia, arthralgia, stomatitis, hyponatremia, abdominal pain, and fatigue.

The efficacies of pembrolizumab for microsatellite instability (MSI)-high solid tumors [68] and neurotrophic tyrosine receptor kinase (NTRK) inhibitors (entrectinib and larotrectinib) for solid tumors with NTRK fusion have also been reported [69,70]. Only a few biliary tract cancer patients were included in these studies, owing to the rarity of these alterations. MSI-high biliary tract cancer was reported in 2.22% and 1.50% of Japanese cholangiocarcinoma and gallbladder cancer patients, respectively [71]. NTRK fusion positivity was reported in only 0.18% of biliary tract cancers [72]. The efficacy and safety of pembrolizumab were evaluated in KEYNOTE-028 and KEYNOTE-158 [73]. Pembrolizumab provides durable antitumor activity in 6–13% of patients with advanced biliary tract cancer regardless of programmed cell death 1 ligand 1 (PD-L1) expression and has manageable toxicity. Other immune checkpoint inhibitors were also evaluated in phase I or II studies involving both naïve and refractory advanced biliary tract cancer [74,75,76,77,78,79]. The results of these studies were promising, and further large-scale evaluation is underway. When using these immune checkpoint inhibitors, appropriate management of immune-related adverse events is required.

In summary, FOLFOX is becoming the standard second-line chemotherapy for refractory cases. The presence of IDH mutations, FGFR fusion/rearrangement and NTRK fusion, as well as MSI status, should be confirmed to consider treatment with relevant inhibitors or immune checkpoint inhibitors where applicable. It is also important to consider participation in clinical studies if molecular-targeted agents matched with identified gene alterations are available.

## 4. Adjuvant Chemotherapy for Resected Biliary Tract Cancer

While surgical resection is regarded as the only treatment with a chance of curing biliary tract cancer, postoperative recurrence can sometimes occur. However, standard adjuvant chemotherapy has not been established to date.

Several phase III studies have been reported on adjuvant chemotherapy for resected biliary tract cancer. The first phase III study evaluated the efficacy of adjuvant chemotherapy of 5-fluorouracil + mitomycin-C versus surgery alone in patients with resected pancreaticobiliary carcinoma [80]. Results indicated that gallbladder carcinoma patients who underwent noncurative resection may derive some benefit from systemic chemotherapy. However, alternative modalities must be developed for patients with carcinomas of the pancreas, bile duct, or ampulla of Vater. Several prospective phase III studies focused on adjuvant chemotherapy for biliary tract cancer were subsequently conducted, as summarized in Table 3.

ESPAC-3 was a phase III study that evaluated the efficacy of adjuvant chemotherapy using 5-fluorouracil + folinic acid or gemcitabine monotherapy against surgery alone [81]. Patients with extrahepatic cholangiocarcinoma and ampullary cancer were enrolled in this study. This study did not show superiority of adjuvant chemotherapy over surgery alone based on an intention-to-treat analysis. However, sensitivity analysis adjusted for prognostic factors showed improved prognosis in both the adjuvant chemotherapy group and the gemcitabine monotherapy group compared to the surgery alone group. BCAT was a phase III study conducted to evaluate the efficacy of adjuvant chemotherapy using gemcitabine against surgery alone [82]. This Japanese study was limited to extrahepatic cholangiocarcinoma patients. Treatment outcomes of surgery alone were extremely good, and no additional benefits of gemcitabine were observed. PRODIGE 12-ACCORD 18 was a French phase III study that compared adjuvant gemcitabine and oxaliplatin combination chemotherapy with surgery alone [83]. All types of biliary tract cancer other than ampullary cancer were included. The efficacy of adjuvant combination chemotherapy was not demonstrated in this negative study. BILCAP was a British phase III study that compared adjuvant capecitabine and surgery alone [84]. While capecitabine monotherapy failed to show improvement based on an intention-to-treat analysis, significant improvement was demonstrated in a per-protocol analysis. The major grade 3/4 adverse events of capecitabine were hand-foot syndrome, diarrhea and fatigue. Because of this promising result, the American Society of Clinical Oncology guideline recommends adjuvant capecitabine monotherapy for resected biliary tract cancer [85].

In summary, capecitabine monotherapy of six months for adjuvant chemotherapy is considered standard treatment for resected biliary tract cancer in Western counties. Until prospective studies show otherwise, surgery alone remains the standard of care in Japan.

## 5. Ongoing Clinical Trials for Biliary Tract Cancer

Currently, effective chemotherapy for biliary tract cancer is extremely limited, and the development of new therapies is urgently needed. There are a large number of ongoing prospective studies for biliary tract cancer [86,87,88,89,90,91]. Based on promising early-phase study results, phase III studies are underway [92,93,94]. A list of major ongoing randomized controlled studies for biliary tract cancer is provided in Table 4. In addition to conventional treatments using cytotoxic agents, a wide variety of drugs such as molecular-targeted agents and immune checkpoint inhibitors are being investigated. Despite the low frequency of genetic alterations, precision medicine with molecular-targeted agents holds promise for selected patients. Umbrella and basket studies are increasingly being conducted, based on the need to build a mechanism to provide drugs suited to each genetic alteration regardless of tumor origin. The efficacy of immunotherapy combined with conventional treatment is also being investigated. In addition, a new large-scale trial for neoadjuvant chemotherapy is underway. Many new therapies that enhance the effectiveness of current regimens have been validated in late-phase clinical trials such as those listed in Table 4. On the other hand, many new drugs have been validated in other, slightly earlier phase clinical trials. It is hoped that such drugs will advance to late-phase clinical trials sooner. Like other cancers, it is also expected that molecular-targeted drugs and immunotherapy that matched cancer genetic characteristics, such as first-line FGFR inhibitors, can produce much better treatment than current standard treatments.

## 6. Conclusions

Figure 1 shows the proposed treatment algorithm of chemotherapy for advanced biliary tract cancer in 2021. It is necessary to arrange this algorithm according to the medical situation in each country.

Biliary tract cancer is considered a population with various genetic alterations. Genetic alterations are often measured before starting second- or third-line chemotherapy only in patients who are able to get enough tissue samples. If the effectiveness of molecular-targeted drugs and immunotherapy based on the characteristics of cancer is shown at first-line setting, it is thought that the trend of investigating genetic alterations from the time of diagnosis will accelerate in the future. In addition, to overcome the problem that biliary tract cancer is sometimes difficult to get enough tissue samples, there are great expectations for liquid biopsy in this field. Furthermore, there is an urgent need to develop more drugs that match genetic alterations and establish a system to deliver the drugs to the matched patients in clinical practice.

While evidence relating to chemotherapy for biliary tract cancer had been limited, numerous clinical studies have been conducted in the last decade and evidence is steadily accumulating. Many large-scale clinical studies are still underway, some of which may lead to improved treatment outcomes going forward.

## Figures and Tables

**Figure 1 jcm-10-03108-f001:**
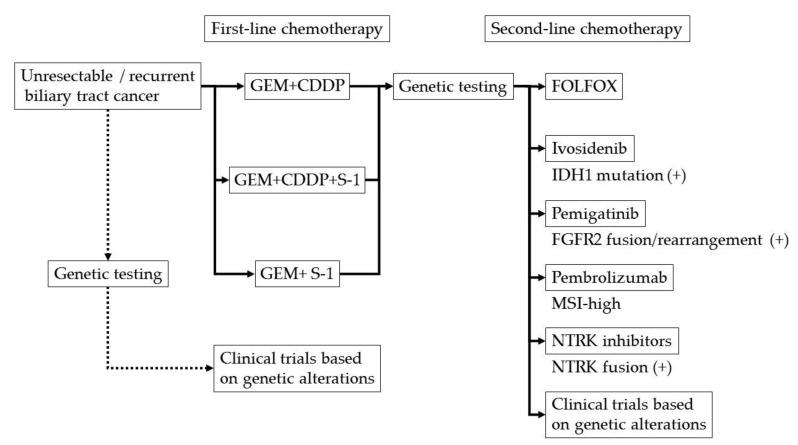
Proposed treatment algorithm of chemotherapy for advanced biliary tract cancer. GEM; gemcitabine, CDDP; cisplatin, FOLFOX; 5-fluorouracil + leucovorin + oxaliplatin, IDH; isocitrate dehydrogenase, FGFR; fibroblast growth factor receptor, MSI; microsatellite instability, NTRK; neurotrophic tyrosine receptor kinase.

**Table 1 jcm-10-03108-t001:** Randomized controlled studies on first-line chemotherapy for advanced biliary tract cancer.

Authors	Year	Regimen	Phase	Result	N	RR	Median PFS	Median OS
Valle et al. [13]	2010	GemCis	3	Positive	204	26.1%	8.0 M	11.7 M
		GEM			206	15.5%	5.0 M	8.1 M
Sharma et al. [10]	2010	GEMOX	3	Positive	26	30.7%	8.5 M	9.5 M
		5FU + FA			28	14.3%	3.5 M	4.6 M
		BSC			27	0%	2.8 M	4.5 M
Lee et al. [17]	2012	GEMOX + Erlotinib	3	Negative	135	29.6%	5.8 M	9.5 M
		GEMOX			133	15.8%	4.2 M	9.5 M
Sharma et al. [16]	2019	GEMOX	3	Negative	119	25.2%	5.0 M	9.0 M
		GemCis			124	23.4%	4.0 M	8.3 M
Morizane et al. [18]	2019	GEM + S-1	3	Positive	179	29.8%	6.8 M	15.1 M
		GemCis			175	32.4%	5.8 M	13.4 M
Sakai et al. [19]	2018	GemCis + S-1	3	Positive	123	41.5%	7.4 M	13.5 M
		GemCis			123	15.0%	5.5 M	12.6 M
Kim et al. [20]	2019	Cape + Oxaliplatin	3	Positive	108	15.7%	5.8 M	10.6 M
		GEMOX			114	24.6%	5.3 M	10.4 M
Phelip et al. [21]	2020	mFOLFIRINOX	2/3	Negative	94	25.0%	6.2 M	11.7 M
		GemCis			96	19.4%	7.4 M	14.3 M
Kang et al. [22]	2012	S-1 + CDDP	rP2	Positive	47	23.8%	5.4 M	9.9 M
		GemCis			49	19.6%	5.7 M	10.1 M
Lee et al. [23]	2015	Cape + CDDP	rP2	Positive	44	27.3%	5.2 M	10.7 M
		GemCis			49	6.1%	3.6 M	8.6 M
Malka et al. [24]	2014	GEMOX + Cmab	rP2	Negative	76	23.1%	6.0 M	11.0 M
		GEMOX			74	29.0%	5.3 M	12.4 M
Chen et al. [25]	2015	GEMOX + Cmab	rP2	Negative	62	27.4%	6.7 M	10.6 M
		GEMOX			60	16.7%	4.1 M	9.8 M
Leone et al. [26]	2016	GEMOX + Pmab	rP2	Negative	45	24.4%	7.7 M	9.5 M
		GEMOX			44	18.2%	5.5 M	9.9 M
Vogel et al. [27]	2018	GemCis + Pmab	rP2	Negative	62	45.2%	6.5 M	12.8 M
		GemCis			28	39.3%	8.3 M	20.1 M
Valle et al. [28]	2015	GemCis + Cediranib	rP2	Negative	62	44.1%	7.7 M	14.1 M
		GemCis			62	18.5%	7.4 M	11.9 M
Moehler et al. [29]	2014	GEM + Sorafenib	rP2	Negative	52	14.3%	3.0 M	8.4 M
		GEM			50	10.0%	4.9 M	11.2 M
Santoro et al. [30]	2015	GEM + Vandetanib	rP2	Negative	58	19.3%	3.8 M	9.5 M
		GEM			56	13.5%	4.9 M	10.2 M
		Vandetanib			59	3.6%	3.5 M	7.6 M
Schnizari et al. [31]	2017	FOLFOX4	rP2	Positive	25	28.0%	5.2 M	13.0 M
		5FU + LV			23	21.7%	2.8 M	7.5 M
Markussen et al. [32]	2020	GEMOX + Cape	rP2	Negative	47	17.0%	5.7 M	8.7 M
		GemCis			49	16.3%	7.3 M	12.0 M
dos Santos et al. [33]	2020	CPT-11 + CDDP	rP2	Positive	24	35%	5.3 M	11.9 M
		GemCis			23	31.8%	7.8 M	9.8 M

N; number, RR; response rate, PFS; progression-free survival, OS; overall survival, M; months, rP2; randomized phase II study, GemCis; gemcitabine + cisplatin, GEM; gemcitabine, GEMOX; gemcitabine + oxaliplatin, 5FU; 5-fluorouracil, FA; folinic acid, BSC; best supportive care, CDDP; cisplatin, Cape; capecitabine, mFOLFIRINOX; modified FOLFIRINOX (5-fluorouracil + leucovorin + irinotecan + oxaliplatin), Cmab; cetuximab, Pmab; panitumumab; FOLFOX; 5-fluorouracil + leucovorin + oxaliplatin, LV; leucovorin, CPT-11; irinotecan.

**Table 2 jcm-10-03108-t002:** Randomized controlled studies on second-line or third-line chemotherapy for advanced biliary tract cancer.

Authors	Year	Regimen	Phase	Result	N	RR	Median PFS	Median OS
Abou-Alfa et al. [49]	2020	Ivosidenib	3	Positive	124	2.4%	2.7 M	10.8 M
		BSC			61	0%	1.4 M	9.7 M
Lamarca et al. [50]	2021	FOLFOX	3	Positive	81	4.9%	4.0 M	6.2 M
		ASC			81	-	-	5.3 M
Jalve et al. [51]	2020	Cape + Varlitinib	2/3	Negative	64	9.4%	2.8 M	7.8 M
		Cape			63	4.8%	2.8 M	7.5 M
Cereda et al. [52]	2016	Cape + MMC	rP2	Negative	29	3.4%	2.3 M	8.1 M
		Cape			28	0%	2.1 M	9.5 M
Zheng et al. [53]	2018	Cape + Irinotecan	rP2	Positive	30	13.3%	3.7 M	10.1 M
		Irinotecan			30	6.7%	2.4 M	7.3 M
Kim et al. [54]	2020	Trametinib	rP2	Negative	24	8.3%	1.4 M	4.3 M
		5FU + LV or Cape			20	10.0%	3.3 M	6.6 M
Demols et al. [55]	2020	Regorafenib	rP2	Positive	33	0%	3.0 M	5.3 M
		BSC			33	0%	1.5 M	5.1 M
Ueno et al. [56]	2021	S-1 + Resminostat	rP2	Negative	50	6.0%	2.9 M	7.8 M
		S-1			51	9.8%	3.0 M	7.5 M
Ramaswamy et al. [57]	2021	Cape + Irinotecan	rP2	Negative	49	6.1%	2.3 M	5.2 M
		Irinotecan			49	0%	3.1 M	6.3 M
Yoo et al. [58]	2021	5FU + LV + nal-IRI	rP2	Positive	88	14.8%	7.1 M	8.6 M
		5FU + LV			86	5.8%	1.4 M	5.5 M

N; number, RR; response rate, PFS; progression-free survival, OS; overall survival, M; months, rP2; randomized phase 2 study, BSC; best supportive care, FOLFOX; 5-fluorouracil + leucovorin + oxaliplatin, ASC; active symptom control, Cape; capecitabine, MMC; mitomycin-C, 5FU; 5-fluorouracil, LV; leucovorin, nal-IRI; nano-liposomal irinotecan.

**Table 3 jcm-10-03108-t003:** Randomized controlled studies of adjuvant chemotherapy for resected biliary tract cancer.

Authors	Year	Biliary Site	Regimen	Phase	Result	N	Median RFS	Median OS
Neoptolemos et al. [81]	2012	EHCC, AC	5FU + FA	3	Marginal	143	23.0 M	38.9 M
			GEM			141	29.1 M	45.7 M
			Surgery alone			144	19.5 M	35.2 M
Ebata et al. [82]	2018	EHCC	GEM	3	Negative	117	36.0 M	62.3 M
			Surgery alone			108	39.9 M	63.8 M
Edeline et al. [83]	2019	ICC, EHCC, GBC	GEMOX	3	Negative	94	30.4 M	75.8 M
			Surgery alone			99	18.5 M	50.8 M
Primrose et al. [84]	2019	ICC, EHCC, GBC	Capecitabine	3	Marginal	223	24.4 M	51.1 M
			Surgery alone			224	17.5 M	36.4 M

N; number, RFS; recurrent-free survival, OS; overall survival, M; months, EHCC; extrahepatic cholangiocarcinoma, AC; ampullary cancer, ICC; intrahepatic cholangiocarcinoma, GBC; gallbladder cancer, 5FU; 5-fluorouracil, FA; folinic acid, GEM; gemcitabine, GEMOX; gemcitabine + oxaliplatin.

**Table 4 jcm-10-03108-t004:** Major ongoing clinical studies for biliary tract cancer.

Regimen	N	Phase	Trial ID
First-line chemotherapy			
NUC-1031 (Acelarin) + CDDP vs. GemCis (NuTide:121)	828	3	NCT04163900
GemCis + Pembrolizumab vs. GemCis (KEYNOTE-966)	788	3	NCT04003636
GemCis + Durvalumab vs. GemCis (TOPAZ-1)	757	3	NCT03875235
Pemigatinib vs. GemCis (FIGHT-302)	432	3	NCT03656536
GEMOX + KN035 vs. GEMOX (KN035-BTC)	390	3	NCT03478488
Infigratinib vs. GemCis (PROOF 301 trial)	384	3	NCT037773302
GemCis + nab-paclitaxel vs. GemCis (SWOG/S1815)	268	3	NCT03768414
Futibatinib vs. GemCis (FOENIX-CCA3)	216	3	NCT04093362
GemCis + Bintrafusp alfa vs. GemCis	512	2/3	NCT04066491
Second-line chemotherapy			
TQB2450 + Anlotinib vs. Cape + Oxaliplatin or Cape + GEM	392	3	NCT04809142
Surufatinib vs. Cape	298	2/3	NCT03873532
Adjuvant chemotherapy			
GemCis vs. Surgery alone or Cape (ACTICCA-1)	781	3	NCT02170090
GEM + Cape vs. Cape (AdBTC-1)	460	3	NCT03779035
S-1 vs. Surgery alone (ASCOT)	350	3	UMIN000011688
Neoadjuvant chemotherapy			
Neoadjuvant & adjuvant GemCis vs. Adjuvant CTx (GAIN)	300	3	NCT03673072
Neoadjuvant GCS vs. Surgery first (NABICAT)	300	3	jRCTs031200388

CDDP; cisplatin, GemCis; gemcitabine + cisplatin, GEMOX; gemcitabine + oxaliplatin, Cape; capecitabine, GEM; gemcitabine, CTx; chemotherapy, GCS; gemcitabine + cisplatin + S-1.

## Data Availability

Not applicable.

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
