# Peer review of "Chemotherapy for Biliary Tract Cancer in 2021"

_jcm, 2021, doi:10.3390/jcm10143108_

Round 1

Reviewer 1 Report

This is a helpful review article on the current standard for systemic therapy for biliary tract cancer.

1.The authors focus very much on “standard therapies”. However, most advances in the recent years have been made with molecular-targeted agents and – to a lesser extent – immunotherapy. I would put a bit more emphasis on this aspect. (a) Which molecular targets should be tested on a tumor sample, e.g., before initiating 2nd-line chemotherapy? A table might be helpful here. (b) Which molecular targeted therapies are already available for BTC?

2.The article provides a good overview on trial results, but does not provide a clear algorithm for clinical practice. It would be helpful to add a figure (or more than one figure) on the 1st, 2nd and 3rd line chemo options for patients (including how to stratify patients for the different options).

3.The article, particularly the tables, focuses a lot on “outcome numbers” (e.g. OS, PFS), but largely neglects toxicity of the different options. This should be added to the article and the tables.

Author Response

Reviewer 1

This is a helpful review article on the current standard for systemic therapy for biliary tract cancer.

  1. The authors focus very much on “standard therapies”. However, most advances in the recent years have been made with molecular-targeted agents and – to a lesser extent – immunotherapy. I would put a bit more emphasis on this aspect. (a) Which molecular targets should be tested on a tumor sample, e.g., before initiating 2nd-line chemotherapy? A table might be helpful here. (b) Which molecular targeted therapies are already available for BTC?

I agree with your comment. I have already mentioned about the question (a) at the last paragraph of second-line chemotherapy. I just want to add the FGFR fusion/rearrangement as below.

“The presence of IDH mutations, FGFR fusion/rearrangement and NTRK fusions as well as MSI status should be confirmed to consider treatment with relevant inhibitors or immune checkpoint inhibitors. It is also important to consider participation in clinical studies if molecular-targeted agents matched with identified gene alterations are available”

As for question (b), I think the agents available for BTC might be different in each countries worldwide. For example, both ivosidenib and FOLFOX are not available in Japan even after the results of phase III trial (ABC-06 and ClarIDHy). Because pemigatinib has been approved in many countries based on only phase II study (not phase III study), I just comment about this point at the last sentence of 3rd paragraph in second-line chemotherapy.

“Based on the results of a phase II study (FIGHT-202) [65], pemigatinib has been approved in many countries for patients with FGFR2 fusion or rearrangement.”

  1. The article provides a good overview on trial results, but does not provide a clear algorithm for clinical practice. It would be helpful to add a figure (or more than one figure) on the 1st, 2nd and 3rd line chemo options for patients (including how to stratify patients for the different options).

Thank you for your constructive opinion. I add the figure at in the part of Conclusions.

  1. The article, particularly the tables, focuses a lot on “outcome numbers” (e.g. OS, PFS), but largely neglects toxicity of the different options. This should be added to the article and the tables.

Thank you for your wonderful suggestion. If we include the information about side effect in the tables, it will become a very complicated table. Moreover, the information of side effect on the many negative regimens are also included, which makes it difficult to see. Therefore, the side effects of the main regimens have been added in the article.

“The major grade 3/4 adverse events of gemcitabine and cisplatin combination chemotherapy were neutropenia and anemia. We also need to pay attention to renal dysfunction and hearing loss.”

“The major grade 3/4 adverse event of gemcitabine and S-1 combination chemotherapy was neutropenia. We also need to pay attention to diarrhea, oral mucositis, maculopapular rash, and skin hyperpigmentation.”

“The major grade 3/4 adverse event of triplet chemotherapy was also neutropenia. This triplet is also needed to pay attention to diarrhea, stomatitis, and rash.”

“The major grade 3/4 adverse events of FOLFOX were neutropenia, fatigue, and catheter-related infection. We also need to pay attention to peripheral neuropathy.”

“The major grade 3/4 adverse events of ivosidenib was reported as ascites.”

“The major grade 3/4 adverse events of pemigatinib were hypophosphatemia, arthralgia, stomatitis, hyponatremia, abdominal pain, and fatigue.”

“When using these immune check point inhibitors, appropriate management of immune-related adverse events are required.”

“The major grade 3/4 adverse events of capecitabine were hand-foot syndrome, diarrhea and fatigue.”

Reviewer 2 Report

In this review the authors summarized the previous and current clinical trials of chemotherapy for biliary tract cancer. The review is well-organized with comprehensive clinical trials cited. The major issue is that the review reads more like a demonstration of evidence without authors' thought. Especially in the ongoing clinical trial section, the authors should include more details, for example: what the clinical trials are about and the rationale of how these clinical trails could benefit the current standard therapy. And for the conclusion part, the authors should also add more discussions, comments and perspectives. 

Author Response

Reviewer 2

In this review the authors summarized the previous and current clinical trials of chemotherapy for biliary tract cancer. The review is well-organized with comprehensive clinical trials cited. The major issue is that the review reads more like a demonstration of evidence without authors' thought. Especially in the ongoing clinical trial section, the authors should include more details, for example: what the clinical trials are about and the rationale of how these clinical trials could benefit the current standard therapy. And for the conclusion part, the authors should also add more discussions, comments and perspectives.

Thank you for your constructive suggestion. I added the sentences at the ongoing clinical trial section and the conclusion section as below.

  1. Ongoing clinical trials for biliary tract cancer

Currently, effective chemotherapy for biliary tract cancer is extremely limited, and the development of new therapies is urgently needed. There are a large number of ongoing prospective studies for biliary tract cancer [86–91]. Based on promising early-phase study results, phase III studies are underway [92–94]. A list of major ongoing randomized controlled studies for biliary tract cancer is provided in Table 4. In addition to conventional treatments using cytotoxic agents, a wide variety of drugs such as molecular-targeted agents and immune checkpoint inhibitors are being investigated. Despite the low frequency of genetic alterations, precision medicine with molecular-targeted agents holds promise for selected patients. Umbrella and basket studies are increasingly being conducted, based on the need to build a mechanism to provide drugs suited to each genetic alteration regardless of tumor origin. Efficacy of immunotherapy combined with conventional treatment is also being investigated. In addition, a new large-scale trial for neoadjuvant chemotherapy is underway. Many new therapies that enhance the effectiveness of current regimens have been validated in late phase clinical trials such as those listed in Table 4. On the other hand, many new drugs have been validated in other, slightly earlier phase clinical trials. It is hoped that such drugs will advance to late phase clinical trials sooner. Like other cancer, it is also expected that molecular-targeted drugs and immunotherapy that matched cancer genetic characteristics, such as first-line FGFR inhibitors, can produce much better treatment than current standard treatments.

Conclusions

Figure 1 shows the proposed treatment algorithm of chemotherapy for advanced biliary tract cancer in 2021. It is necessary to arrange this algorithm according to the medical situation in each countries.

              Biliary tract cancer is considered as a population with various genetic alterations. Genetic alterations are often measured before starting second- or third-line chemotherapy only in patients who are able to get enough tissue samples. If the effectiveness of molecular-targeted drugs and immunotherapy based on the characteristics of cancer is shown at first-line setting, it is thought that the trend of investigating genetic alterations from the time of diagnosis will accelerate in the future. In addition, to overcome the problem that biliary tract cancer is sometimes difficult to get enough tissue samples, there are great expectations for liquid biopsy in this field. Furthermore, there is an urgent need to develop more drugs that match genetic alterations and establish a system to deliver the drugs to the matched patients in clinical practice.

While evidence relating to chemotherapy for biliary tract cancer had been limited, numerous clinical studies have been conducted in the last decade and evidence is steadily accumulating. Many large-scale clinical studies are still underway, some of which may lead to improved treatment outcomes going forward.
